# Differential ABC transporter expression during hematopoiesis contributes to neutrophil-biased toxicity of Aurora kinase inhibitors

David B. Chou [1,2,9], Brooke A. Furlong [1,9], Ryan R. Posey[1,9], Christos Kyprianou [1], Lucy R. O'Sullivan[1], Rhiannon David[3], Suzanne J. Randle[3], Urszula M. Polanska[4], Jon Travers[4], Jelena Urosevic[4], John N. Hutchinson[5], Jianwei Che[6], Anna M. Howley[1], Robert P. Hasserjian[2], Rachelle Prantil-Baun[1] & Donald E. Ingber [1,7,8] ✉

Drug-induced cytopenias are a prevalent and significant issue that worsens clinical outcomes and hinders the effective treatment of cancer. While reductions in blood cell numbers are classically associated with traditional cytotoxic chemotherapies, they also occur with newer targeted small molecules and the factors that determine the hematotoxicity profiles of oncologic drugs are not fully understood. Here, we explore why some Aurora kinase inhibitors cause preferential neutropenia. By studying drug responses of healthy human hematopoietic cells in vitro and analyzing existing gene expression datasets, we provide evidence that the enhanced vulnerability of neutrophil-lineage cells to Aurora kinase inhibition is caused by early developmental changes in ATP-binding cassette (ABC) transporter expression. These data show that hematopoietic cell-intrinsic expression of ABC transporters may be an important factor that determines how some Aurora kinase inhibitors affect the bone marrow.

Drug-induced cytopenias are a significant cause of morbidity and mortality among patients treated for cancer[1,2]. These reductions in the number of blood cells are often dose limiting and prevent patients from receiving optimal therapy for their disease. As such, hematologic parameters are routinely assessed in patients undergoing cancer treatment and rates of neutropenia, anemia, and thrombocytopenia are measured whenever novel cancer therapies are tested in clinical trials. Furthermore, each individual drug often exhibits lineage-bias in its side effect profile, with certain types of cytopenias more commonly seen than others.

One example is AZD2811, a highly potent inhibitor of Aurora kinase B that has been tested in clinical trials (as the prodrug AZD1152) for use in a variety of solid and hematologic cancers[3–5]. In multiple phase I trials of this drug, a significant fraction of patients experienced severe neutropenia that was dose limiting[3–5], whereas severe anemia and thrombocytopenia were rare or non-existent. As the mechanism of action of AZD2811 is based on mitotic inhibition, it is unclear how this disparity in the rates of cytopenias for different hematopoietic lineages arises.

Proteins in the Aurora kinase (AURK) family, consisting of AURKA, AURKB, and AURKC, are key regulators of the onset and progression of

[1]Wyss Institute for Biologically Inspired Engineering at Harvard University, Boston, MA, USA. [2]Department of Pathology, Massachusetts General Hospital, Boston, MA, USA. [3]Safety Sciences, Clinical Pharmacology and Safety Sciences, R&D, AstraZeneca, Cambridge, UK. [4]Early Oncology, R&D, AstraZeneca, Cambridge, UK. [5]Harvard Chan Bioinformatics Core, Boston, MA, USA. [6]Department of Cancer Biology, Dana-Farber Cancer Institute, and Department of Biological Chemistry & Molecular Pharmacology, Harvard Medical School, Boston, MA, USA. [7]Vascular Biology Program and Department of Surgery, Boston Children's Hospital and Harvard Medical School, Boston, MA, USA. [8]Harvard John A. Paulson School of Engineering and Applied Sciences, Boston, MA, USA. [9]These authors contributed equally: David B. Chou, Brooke A. Furlong, Ryan R. Posey. ✉e-mail: don.ingber@wyss.harvard.edu

cell division[6,7], which is critical for both normal and malignant cell growth. Multiple inhibitors targeting the AURK family have been synthesized and are in active clinical development for various cancers[8,9]. AZD2811 targets AURKB, which phosphorylates histone H3 at serine 10 (a classic epigenetic mark of mitotic chromosomes) and regulates chromosome segregation and cytokinesis[6,7]. AURKB inhibition results in loss of phosphorylated histone H3 serine 10, mitotic failure with polyploidy, and cell death in multiple cancer cell types[10–12].

This mechanism of action would seem to be cell-type independent, so why is the neutrophil lineage preferentially affected in patients treated with this drug? In this study, we address this question and show that this neutropenia bias is shared by multiple other AURK inhibitors. We then recapitulate the neutrophil sensitivity in vitro and demonstrate that the neutrophil lineage is susceptible to AZD2811 at lower concentrations and recovers more slowly upon drug withdrawal than non-neutrophil lineage cells. We find that neutrophil sensitivity to AZD2811 develops early during lineage commitment and this correlates with loss of ABC transporter expression. Finally, we demonstrate that blocking ABC transporter function equalizes the response of neutrophil and non-neutrophil lineages to multiple AURK inhibitors, including AZD2811.

## Results

### Multiple Aurora kinase inhibitors preferentially induce neutropenia in patients

Severe neutropenia is the dominant dose-limiting side effect for patients treated with AZD2811 (administered as the prodrug AZD1152), whereas severe anemia is rare and severe thrombocytopenia is not observed[3,5]. To determine whether this lineage-specific hematotoxicity occurs with other AURK inhibitors besides AZD2811, we searched the literature for clinical trials that tested AURK inhibitors as single agents and aggregated the observed rates of cytopenias and their severity (Table 1). We found data for 8 additional drugs (danusertib, tozasertib, AMG 900, AT9283, MSC1992371A, PF-03814735, ENMD-2076, and MLN8237). Of note, while AZD2811 is highly specific for AURKB, these other drugs have reported activity against other AURK family

members and occasionally other kinases as well. Nevertheless, we detected the same neutrophil-biased side effect profile across all 9 different drugs tested in 26 clinical trials involving over 1600 total patients. Furthermore, neutropenia was determined to be a dose-limiting toxicity for all of these AURK inhibitors.

Given these findings, we next sought to determine whether other targeted small molecule cancer drugs exhibit similar neutropenia-based toxicity. To address this question, we analyzed the set of FDA-approved small molecule cancer therapies, which is comprised of over 80 chemically diverse compounds for which robust and extensive patient cytopenia data are available. After removing studies of patients with acute leukemias and myeloid neoplasms (which often present as severe cytopenias at baseline and have significantly altered erythroid, myeloid, and megakaryocytic development) as well as studies where the drug of interest was paired with other myelosuppressive drugs, 61 drugs remained for which hematotoxicity data were available. When the rates of severe cytopenias (CTCAE grade 3 or higher) for these drugs were examined, 29 of the 61 FDA-approved targeted small molecule cancer therapeutics were found to induce severe cytopenias in at least 5% of patients. However, those cytopenias were not neutropenia-biased as a whole. In comparison, AURK inhibitors exhibited a significant preference for inducing neutropenia, with multiple drugs causing this toxicity in more than 40% of patients (Supplementary Fig. 1a, b and Supplementary Table in Source Data). This analysis demonstrated that AURK inhibitor drugs are significantly more likely to cause neutrophil-biased cytopenias (8 out of 9 drugs). This behavior is not influenced by the AURK isoform selectivity of the inhibitors as neutropenias were induced by both AURKA and AURKB selective inhibitors (Supplementary Fig. 2).

We also analyzed the chemical structures of the AURK inhibitors and the relevant filtered list of FDA-approved compounds described above to identify if there might be structural features that associate with induction of neutropenia. To do this, we performed molecular fingerprinting analysis using extended-connectivity fingerprint of diameter 6 (ECFP_6), which calculates a numerical fingerprint for each drug based upon all possible substructures within the molecule with a maximum radius of 3 bonds (and thus a diameter of 6)[13,14]. This is a widely used fingerprinting method that has proven utility in a variety of scientific applications[14]. Clustering of the molecules based on their ECFP_6 fingerprint showed that the AURK inhibitors did not cluster separately from the FDA-approved drugs (Supplementary Fig. 3a). Furthermore, neutropenia-biased drugs did not cluster separately from drugs without preferential neutropenia (Supplementary Fig. 3a). Thus, according to their ECFP_6 representation, all of these compounds distribute over the same region of chemical space. Analysis of the overall lipophilicity of the compounds also showed no significant difference between AURK inhibitors and the FDA-approved compounds or between drugs with and without a neutropenia-bias (Supplementary Fig. 3b, c). Taken together, these findings show the neutrophil-biased toxicity profile of AURK inhibitors is not a general property of small molecule cancer drugs and that this neutropenia-bias occurs in the context of significant chemical structural diversity.

### AZD2811 causes neutrophil-specific toxicity in vitro

To investigate why neutropenia is the dominant cytopenia experienced by patients treated with AURK inhibitors, we sought to understand this phenomenon in vitro. We previously showed that treating suspension cultures of human hematopoietic cells with AZD2811 at a clinically relevant concentration (42 nM) is able to recapitulate aspects of its neutrophil-biased toxicity profile[15]. Here, we used the same system where CD34[+] hematopoietic progenitors are cultured for 10 days with GCSF and erythropoietin which supports neutrophil and erythroid differentiation, followed by 48 h of treatment with 42 nM AZD2811. We again observed that hematopoietic toxicity was restricted to the neutrophil lineage (Fig. 1a).

## Table 1 | Patients treated with existing Aurora kinase inhibitors experience neutropenia-biased hematotoxicity

| Neutropenia is a dose-limiting toxicity for all listed AURK inhibitors | | | | | | |
|---|---|---|---|---|---|---|
| Inhibitor | Trials | Pts | % Neut | % An | % Thr | PMID |
| AZD1152 | 2 | 94 | 40.4 | 2.1 | 0 | 20924078 |
| | | | | | | 22661287 |
| Danusertib | 4 | 390 | 62.6 | 4.6 | 1.5 | 19770380 |
| | | | | | | 19825950 |
| | | | | | | 22928785 |
| | | | | | | 25488684 |
| Tozasertib | 1 | 27 | 18.5 | 3.7 | 0 | 20386909 |
| AMG 900 | 1 | 105 | 41.9 | 26.7 | 12.4 | 29980894 |
| AT9283 | 3 | 108 | 21.3 | 5.6 | 0.9 | 22015452 |
| | | | | | | 24072436 |
| | | | | | | 25370467 |
| MSC1992371A | 1 | 92 | 17.4 | 8.7 | 3.3 | 23832397 |
| PF-03814735 | 1 | 57 | 21.1 | 0 | 5.3 | 21852114 |
| ENMD-2076 | 2 | 131 | 4.6 | 0.8 | 0.8 | 21131552 |
| | | | | | | 22921155 |
| MLN8237 | 11 | 693 | 36.7 | 11.5 | 8.7 | See legend |

Overall percentages of patients experiencing severe (CTCAE grade 3 or higher) neutropenia (% Neut), anemia (% An), and thrombocytopenia (% Thr) were calculated from published clinical trials of AURK inhibitors, excluding studies of patients with hematologic neoplasms. PMIDs for the MLN8237 studies are: 22753585, 22767670, 22772063, 22988055, 24879333, 25728526, 26084989, 26873642, 27502708, 28094040, 30232224.

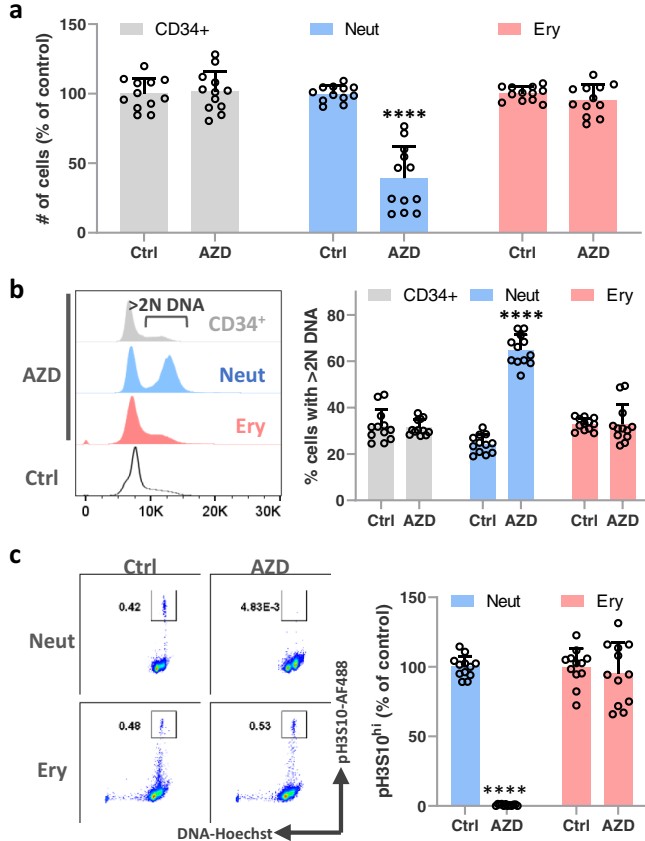

**Fig. 1 | AZD2811 causes neutrophil-specific toxicity in vitro.** Human CD34[+] progenitor cells were cultured for 10 days and then treated for 48 h with AZD2811 at the clinically relevant concentration of 42 nM or left untreated. **a** Cells were harvested and analyzed by flow cytometry to measure cell numbers. **b** DNA content was measured via Hoechst 33342 staining. Representative flow histograms are shown for CD34[+] progenitor, neutrophil lineage, and erythroid cells in the presence of AZD2811 as well as bulk untreated cells. Bar graphs display quantitation of the percentage of cells with >2 N DNA content. **c** Phosphorylation of histone H3 at serine 10 (pH3S10) was measured by intracellular staining and the % of cells with bright pH3S10 staining was quantified relative to controls without AZD2811. Representative flow cytometry plots are shown alongside quantitative bar graphs. Mean and SD are shown; $n = 12$, from 4 independent experiments. ****$p < 0.0001$, two-sided $t$ test. Source data are provided in the Source Data file.

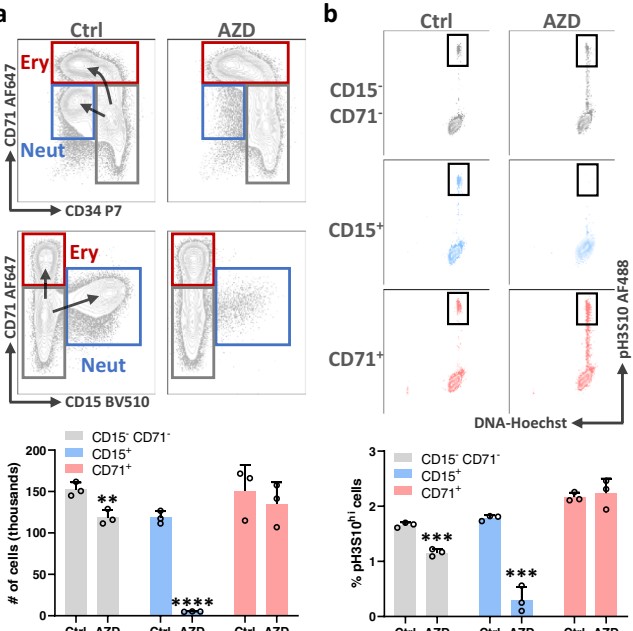

**Fig. 2 | Neutrophil sensitivity to AZD2811 is acquired early in hematopoietic differentiation.** Human CD34[+] progenitor cells were cultured for 4 days and then treated for 48 h with AZD2811 at the clinically relevant concentration of 42 nM or left untreated. **a** Cells were harvested and analyzed by flow cytometry to measure numbers of early erythroid (CD71[bright] CD15[-]; red), early neutrophil (CD15[+] CD34[-]; blue), and remaining CD15[-] CD71[-] CD34[+] progenitor cells (gray). Representative flow plots showing the various populations are shown along with quantitative bar graphs. Mean and SD are shown; $n = 3$, data are from pooled replicates due to low numbers at early timepoint, from 2 independent experiments. **$p = 0.009$, ****$p < 0.0001$, two-sided $t$ test. **b** pH3S10 was measured by intracellular staining and flow cytometry in the 3 different cell populations. Representative flow plots are shown along with bar graphs quantitating the percentage of pH3S10[bright] cells (gated as shown). Mean and SD are shown; $n = 3$, data are from pooled replicates due to low numbers at early timepoint, from 2 independent experiments. ***$p = 0.0004$ for both indicated comparisons, two-sided $t$ test. Source data are provided in the Source Data file.

We then sought to measure directly the ability of AZD2811 to inhibit AURKB in the neutrophil, CD34[+] progenitor, and erythroid cell populations. To do this, we analyzed cell cycle status by quantifying DNA content and assessed phosphorylation of histone H3 at serine 10 (pH3S10), which is the direct target of AURKB and hence widely used to measure AURKB activity. Neutrophil lineage cells treated with AZD2811 showed almost complete loss of pH3S10 and a significant increase in the percentage of cells that replicated their DNA (>2 N DNA content), but were unable to divide (Fig. 1b, c). However, CD34[+] and erythroid lineage cells in the same AZD2811-treated cultures showed similar levels of pH3S10 to control cultures and no change in the fraction of cells with >2 N DNA (Fig. 1b, c). These data demonstrate that while CD34[+] progenitor and erythroid cells were actively cycling and thus should have been susceptible to AURKB inhibition, AZD2811 did not suppress AURKB activity in these cells.

## Neutrophil sensitivity to AZD2811 is acquired early in hematopoietic differentiation

The neutrophils in our cultures arise from CD34[+] progenitors, suggesting that resistance to AZD2811 is lost as the progenitors differentiate toward neutrophils. To determine when this occurs, we initiated the same 48 h AZD2811 treatment on day 2 of culture instead of day 10. When analyzed on day 4, control cultures showed that a majority of cells are just beginning to differentiate into CD71[high] erythroid or CD15[+] neutrophil lineage cells while a significant fraction remain as CD15[-]CD71[-]CD34[+] progenitors (Fig. 2a). AZD2811 treatment largely eliminated the early CD15[+] neutrophil lineage population and the small number of CD15[+] cells present had lost pH3S10 (Fig. 2a, b). As CD15 begins to be expressed in the earliest known unilineage neutrophil progenitor[16,17], these data suggest that sensitivity to AZD2811 is acquired before lineage commitment during neutrophil development.

While the numbers of CD71[high] erythroid and CD15[-]CD71[-] CD34[+] progenitor cells were largely unchanged by AZD2811 (Fig. 2a), we observed that the phosphorylation profile of histone H3 serine 10 in these cells was altered by the drug treatment. In control culture conditions, cells contain very little pH3S10 unless they are actively undergoing mitosis when they have high levels of pH3S10, and this results in a distinct population of pH3S10 bright cells (Fig. 2b). In the presence of AZD2811, however, a continuum of pH3S10 from low to high levels of expression was seen (Fig. 2b). These data indicate that AZD2811 was partially active against erythroid and progenitor cells, but the cells were somehow able to overcome its effect on cell proliferation.

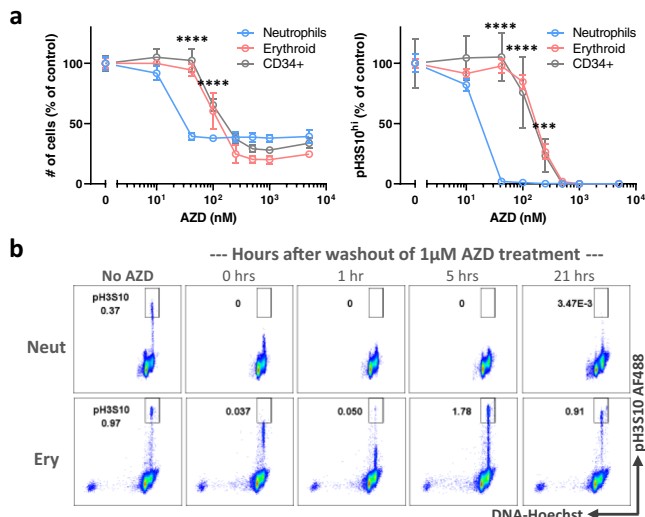

**Fig. 3 | Increased sensitivity of neutrophils to AZD2811 is accompanied by impaired recovery after drug withdrawal. a** Human CD34[+] progenitor cells were cultured for 10 days and then treated for 48 h with AZD2811 at various concentrations or DMSO vehicle. Cells were harvested and analyzed by flow cytometry to measure cell numbers and the percentage of pH3S10[bright] cells. Mean and SD are shown; $n = 6$ for each concentration except $n = 3$ at highest concentration (5 μM), from 2 independent experiments. ***$p = 0.0001$ (erythroid) and .0004 (CD34+), ****$p < 0.0001$, 2-way ANOVA of erythroid and CD34[+] cells compared against neutrophils at AZD2811 various concentrations. **b** Human CD34[+] progenitor cells were cultured for 12 days and then treated for 2 h with 1 μM AZD2811 or left untreated. AZD2811 was then washed out and cells were analyzed at the indicated timepoints after washout by intracellular flow cytometry for pH3S10. Representative flow plots are shown; $n = 4$-6 except 21 h timepoint has 2 data points, from 3 independent experiments. Source data are provided in the Source Data file.

## Increased neutrophil sensitivity to AZD2811 is accompanied by impaired recovery after withdrawal

The prior results suggested that while the clinically relevant concentration of AZD2811 we used was effective in neutrophil lineage cells, higher concentrations might overcome the resistance of the non-neutrophil lineage cells. This was confirmed when we treated 10 day hematopoietic cultures with a range of AZD2811 concentrations for 48 h (Fig. 3a). We now observed effects in all cell types and quantification of pH3S10 levels and cell numbers at day 12 revealed that neutrophil lineage cells were between one half-log to one log more sensitive to AZD2811 than the erythroid and CD34[+] progenitor cells.

This differential sensitivity to AZD2811 implied that various cell lineages also might recover differently after drug withdrawal. To test this hypothesis, we treated 12 day cultures with a 2 h pulse of 1 μM AZD2811, which is a high enough concentration to affect all cell types in our cultures (Fig. 3a). After washing the drug out, we found that pH3S10 levels continued to decrease for about an hour after drug withdrawal in both neutrophil and erythroid lineage cells (Fig. 3b). Erythroid cells, however, proceeded to recover significant AURKB activity within the next 4 h and returned to control pH3S10 levels by the next day. In contrast, neutrophil cells in the same cultures had little detectable histone H3 phosphorylation even after one day in the absence of drug (Fig. 3b).

## Differential expression of ABC transporters in developing neutrophils, erythroid cells and CD34[+] cells

Based on the data above, we hypothesized that CD34[+] progenitors and erythroid cells might be lowering intracellular concentrations of AZD2811 in a way that neutrophils do not, for example, by pumping the drug out of the cell. The ATP-binding cassette (ABC) transporter family in humans comprises 48 different proteins with a variety of cellular

transport functions, which include the ability to transport drugs out of the cell[18,19]. In particular, ABCB1 (MDR1/P-gp), ABCC1 (MRP1), and ABCG2 (BCRP) are key cellular drug efflux pumps and important mediators of drug resistance (hereafter referred to as B1, C1, and G2, respectively)[20–25]. Thus, we performed qPCR analysis to measure the expression of these efflux pumps in CD34[+] progenitors, neutrophils, and erythroid cells in our cultures. We found that expression of B1, C1, and G2 all decreased as CD34[+] progenitor cells differentiated into neutrophils (Fig. 4a). In contrast, erythroid differentiation resulted in decreased expression of B1 and C1 but increased expression of G2 (Fig. 4a).

We then sought to verify our in vitro findings by assessing the expression of these transporters during human hematopoietic differentiation in vivo. Analysis of previously published gene expression microarray data on freshly isolated human blood cells at various stages of differentiation[26] confirmed that neutrophil development is associated with decreased or low expression of all 3 genes (Fig. 4b). Erythroid development, on the other hand, is accompanied by retention of C1 during initial differentiation and significant upregulation of G2 at later stages (Fig. 4b). Importantly, analysis of a separate microarray dataset[27] showed similar results (Supplementary Fig. 4). These data supported our hypothesis that non-neutrophil lineage cells reduce intracellular AZD2811 concentration by pumping drug out of the cell while neutrophils do not.

## Differences in hematopoietic cell sensitivity to AZD2811 depend on ABC transporter function

To establish whether differences in ABC transporter function underlie the differential drug sensitivity between neutrophil and non-neutrophil lineage cells, we used specific small molecule inhibitors of B1, C1, and G2 to block their functions individually and together. Hematopoietic cultures were treated for 48 h with 42 nM AZD2811 starting on day 10 as before, but this time in the presence of inhibitors against B1 (0.5 μM zosuquidar), C1 (50 μM MK-571), and G2 (0.5 μM Ko143) as per previously published studies[18,28–30] (Fig. 5a). Neutrophil lineage cells showed complete loss of pH3S10 upon AZD2811 treatment regardless of whether ABC transporter inhibitors were present. Erythroid cells, in contrast, were not affected by AZD2811 alone or in combination with B1 inhibition, but were sensitive when treated with AZD2811 combined with C1 and/or G2 inhibitors (Fig. 5a). A small population of cells with >4 N DNA is also more clearly seen in these flow plots (Fig. 5a; note that the x-axis is linear as opposed to log scale to better highlight the effect of blocking ABC transporter function). This subpopulation represents polyploid cells with >4 N DNA content which is a previously reported and expected consequence of mitotic inhibition[10]. Similar to the effects on pH3S10, this population appears in the neutrophil lineage upon treatment with AZD2811, but not in erythroid cells unless ABC transporter function is also blocked.

We also observed a similar pattern when we quantified cell numbers. Neutrophil numbers decreased upon AZD2811 treatment and were minimally affected by ABC transporter inhibitors while erythroid cell numbers were markedly reduced when AZD2811 was combined with inhibition of C1 and/or G2 (Fig. 5b). Furthermore, CD34[+] cell numbers only decreased when AZD2811 was combined with inhibition of B1 (Fig. 5b). The correspondence between CD34[+] progenitors and B1, and between erythroid cells and both C1 and G2, align with the gene expression profiles for these transporters during human hematopoietic development in vivo (Fig. 4). Importantly, treatment with the ABC transporter inhibitors without AZD2811 had no effect on pH3S10 or cell numbers (Supplementary Fig. 5). Taken together, these data demonstrate that different levels of expression of specific ABC transporters during hematopoietic development are responsible for the differences in sensitivity of neutrophil and non-neutrophil lineage cells to AURKB inhibition by AZD2811.

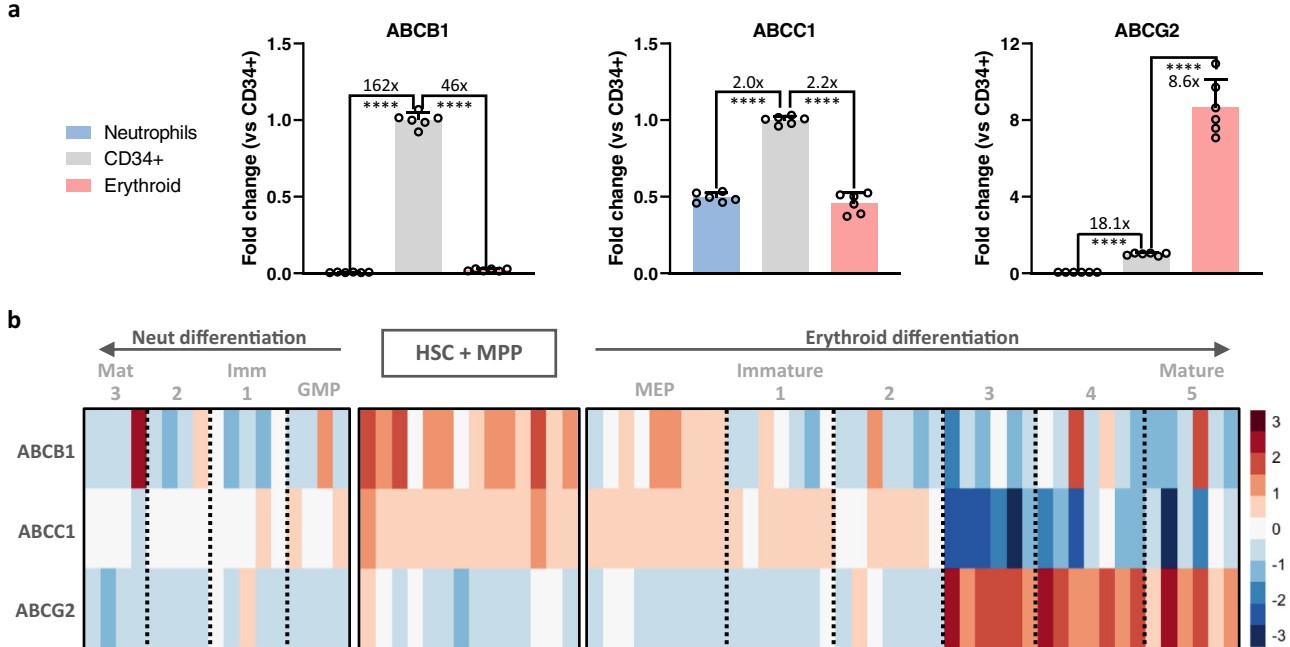

**Fig. 4 | Developing neutrophils lose ABC transporters while developing erythroid cells switch ABC transporters. a** Neutrophil, CD34+ progenitor, and erythroid populations were FACS sorted from 10 day hematopoietic cultures and quantitative real-time PCR was used to measure relative expression of B1, C1, and G2. Bar graphs display fold change normalized to GAPDH and the CD34+ progenitor population. Mean and SD are shown; *n* = 6, from 2 independent experiments. ****p < 0.0001, 1-way ANOVA of normalized cycle threshold counts for each gene with post-hoc analysis via Dunnett's test to compare erythroid and neutrophil cells

against CD34+ cells (control group). **b** Microarray data on developing human hematopoietic cells[26] was analyzed for relative expression of ABCB1, ABCC1, and ABCG2 using the previously defined cell populations as shown. HSC + MPP (lineage⁻CD34+CD38⁻); GMP (CD34+CD38+CD123loCD45RA+); Neut 1 (SSChiCD11b⁻CD16⁻); Neut 2 (SSChiCD11b+CD16⁻); Neut 3 (SSChiCD11b+CD16+); MEP (CD34+CD38+CD123⁻CD45RA⁻); Ery 1 (CD34+CD71+CD235a⁻); Ery 2 (CD34⁻CD71+CD235a⁻); Ery 3 (CD34⁻CD71+CD235a+); Ery 4 (CD34⁻CD71loCD235a+); Ery 5 (CD34⁻CD71⁻CD235a+). Source data are provided in the Source Data file.

## ABC transporter function underlies neutrophil-biased hematotoxicity for multiple AURK inhibitors

To see if our findings were applicable to other AURK inhibitor drugs, we assessed the behavior of 2 additional representative members of the AURKi family in our in vitro cultures. Danusertib (aka PHA-739358) and tozasertib (aka MK-0457 or VX-680) are small molecule kinase inhibitors with activity against all AURKs, though they differ from AZD2811 in their selectivity for AURKB vs AURKA (Supplementary Fig. 2) and also have activity against other kinases (e.g., Abl)[31–38]. However, in clinical trials, they showed a similar neutropenic profile to AZD2811 (Table 1). As with AZD2811, we treated hematopoietic cultures with each compound for 48 h beginning on day 10 at a range of doses and observed a significant loss of neutrophil lineage cells at concentrations that left erythroid and CD34+ progenitor cell numbers unchanged (Supplementary Fig. 6). And similar to the results seen with AZD2811, inhibition of ABC transporter function in the presence of either drug resulted in sensitization of erythroid and CD34+ progenitor cells (Fig. 6a).

Overall, our data are consistent with a model (Fig. 6b) in which CD34+ progenitor cells are resistant to some AURK inhibitors due to expression of the drug efflux transporter B1. As the progenitors differentiate into neutrophils, B1 expression is lost at an early stage of development and all neutrophil lineage cells are rendered sensitive to those AURK inhibitors. In contrast, erythroid differentiation results in loss of B1 but retention of C1 in early erythropoiesis and gain of G2 at later stages, allowing erythroid cells to remain resistant to the same small molecule drugs. This differential expression of drug efflux transporters may contribute to the strong neutropenia bias in the hematotoxic profile of multiple existing Aurora kinase inhibitors.

## Discussion

Our data suggest that changes in ABC transporter expression within developing hematopoietic cells can significantly influence the specific cytopenias that are experienced by patients receiving cancer therapies, which may help to explain why neutropenia is a dominant dose-limiting toxicity for AURK inhibitors. While we focus on CD34 + progenitor, neutrophil lineage, and erythroid lineage cells in our cultures, microarray gene expression profiling shows that megakaryocytes retain ABCB1 expression (Supplementary Fig. 4), consistent with our proposed explanation for the lower incidence of non-neutrophil cytopenias. Our findings also agree with published data showing that human CD34+ progenitors express functional ABC transporters, including B1, and that neutrophils and monocytes are deficient in functional B1 when compared to other differentiated leukocytes in human peripheral blood[39–44]. We note that both neutrophils and monocytes arise from the granulocyte-monocyte progenitor, suggesting that B1 expression during differentiation is downregulated during or before generation of that common progenitor cell. Our in vitro culture data confirm loss of B1 function before unilineage neutrophil commitment and we present gene expression data showing B1 expression is indeed reduced in granulocyte-monocyte progenitors.

The changing expression patterns of ABC transporters during hematopoietic development raise questions about what physiologic roles these proteins might have in normal hematopoietic cell biology. It has been proposed that ABC transporters in stem cells may protect against naturally occurring toxins but a definitive functional role has not been identified[39,41,45]; for example, B1 knockout mice lack an obvious hematopoietic phenotype[46,47]. One of the clearest links between ABC transporter expression and hematopoietic cell-specific function is that developing erythroid cells in mice upregulate G2 and this may control intracellular protoporphyrin IX levels, an

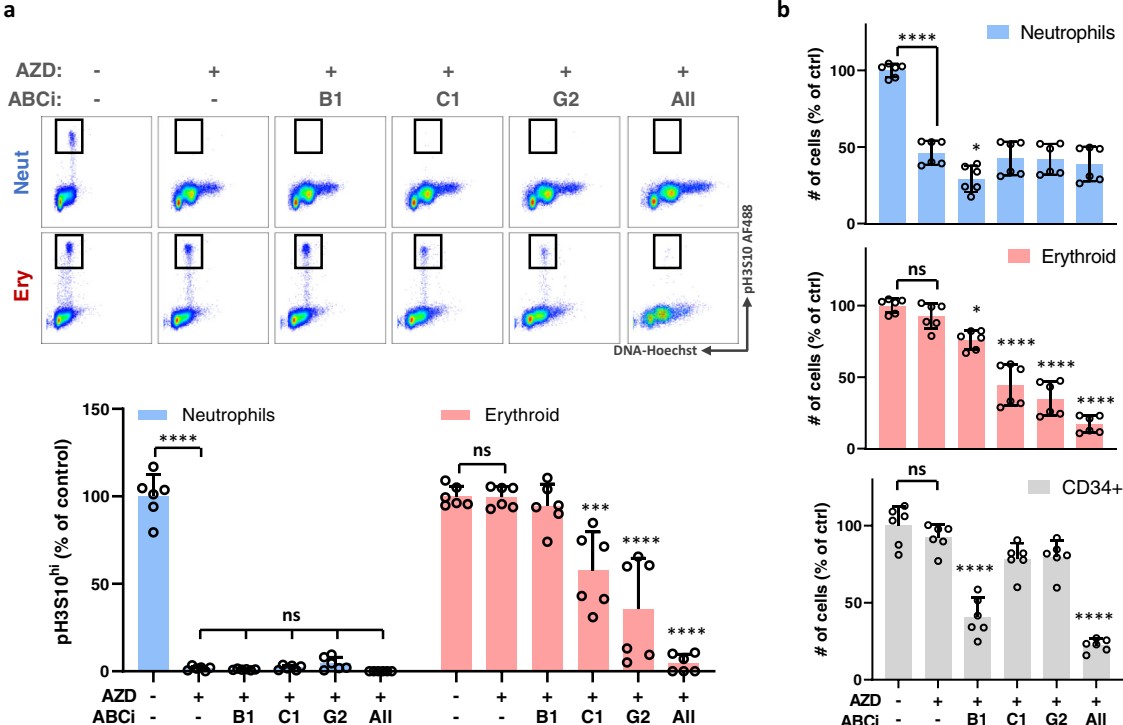

**Fig. 5 | Differences in hematopoietic cell sensitivity to AZD2811 are dependent on ABC transporter function.** Human CD34+ progenitor cells were cultured for 10 days and then treated for 48 h with 42 nM AZD2811 or vehicle control. ABC transporter inhibitors (ABCi) targeting B1 (0.5 μM zosuquidar), C1 (50 μM MK-571), and G2 (0.5 μM Ko143) were also added at the time of drug treatment as indicated. **a** Cells were harvested and pH3S10 was measured by flow cytometry. Representative flow plots are shown along with bar graphs quantitating the percentage of pH3S10bright cells. Mean and SD are shown; $n = 6$, from 2 independent experiments.

$***p = 0.0005$, $****p < 0.0001$, 1-way ANOVA with post-hoc analysis via Dunnett's test to compare the AZD-only group against untreated or AZD + ABCi-treated groups. **b** Cell numbers were measured by flow cytometry. Mean and SD are shown; $n = 6$, from 2 independent experiments. $*p = 0.0137$ (neutrophils), $*p = 0.0210$ (erythroid), $****p < 0.0001$, ns (not significant), 1-way ANOVA with post-hoc analysis via Dunnett's test to compare the AZD-only group against untreated or AZD + ABCi-treated groups. Source data are provided in the Source Data file.

intermediate in the biosynthesis of heme molecules[45]. However, mice lacking ABCG2 still have normal hematopoietic development (including erythropoiesis)[48,49]. The reason for loss of ABC transporter function during the development of specific hematopoietic lineages is also unclear. Our review of the literature did not find any studies that established whether ABC transporter downregulation is necessary for the proper function of neutrophils or monocytes.

Given that the neutropenia bias exhibited by some AURK inhibitors may be ABC transporter-dependent, we asked if there are common structural features that predispose them to be substrates for drug efflux pumps. Additionally, we investigated if there are structural features within a relevant filtered list of FDA-approved targeted cancer therapies that are associated with a neutropenia-biased hematotoxicity profile. Our cheminformatics analysis did not find supporting evidence for either hypothesis. Indeed, a significant body of research has proven that these pumps are capable of effluxing a structurally diverse array of molecules and identifying ABC transporter substrates remains a difficult problem in drug development that ultimately relies on empirical methods more than in silico prediction[20,50–52]. We also note that a general neutropenia-bias is not present in the filtered set of FDA-approved small molecule cancer drugs for which we have high quality drug-specific cytopenia data. Since neutrophils and non-neutrophil lineage cells are likely to differentially efflux these other drugs, this suggests the neutrophil sensitivity to AURK inhibitors may be partly attributable to their specific mechanism of action, though it is unclear why mitotic blockade by AURK inhibition would be cell-type specific.

Clinical trials show that patients receiving AURK inhibitors can experience cytopenias in non-neutrophil lineages, although they occur at significantly lower frequencies (Supplementary Figs. 1 and 2). The lower but non-zero incidence of these other cytopenias aligns with our findings as we demonstrate how sufficiently high concentrations of these drugs can overwhelm the drug efflux capacity of developing non-neutrophil hematopoietic cells. In clinical practice, the dosing regimens used for AURK inhibitors may expose developing bone marrow cells to concentrations that reach those levels and this will vary significantly between individuals in a manner that is dependent on each patient's particular pharmacokinetic profile. Aside from differences in bodyweight and liver/kidney function, patients may be taking other medications that interfere with ABC transporter function, making those patients more susceptible to other hematotoxicities. We also note that patients may have ABC transporter polymorphisms, which represents a potential precision medicine opportunity in which those polymorphisms might inform AURK inhibitor use or dosing. Thus, we acknowledge the cytopenias observed in patients are caused by a combination of drug-specific, cell-specific, and patient-specific factors and that our results provide insight into one aspect of a larger complex set of interactions.

Finally, we believe that the findings in this study have potential implications for drug development. Altering the chemical structure of candidate drugs and their active metabolites to be better substrates for B1, C1, and G2 may avoid or reduce hematotoxicity in a lineage-dependent manner. As cancer cells can use these same drug efflux transporters, this may be counterproductive in some oncologic settings but it might be useful for non-oncologic drug development. The fact that certain drugs preferentially affect neutrophils over other blood cells could be a useful property if neutrophils are the intended target cell population. For example, neutrophils are important for the

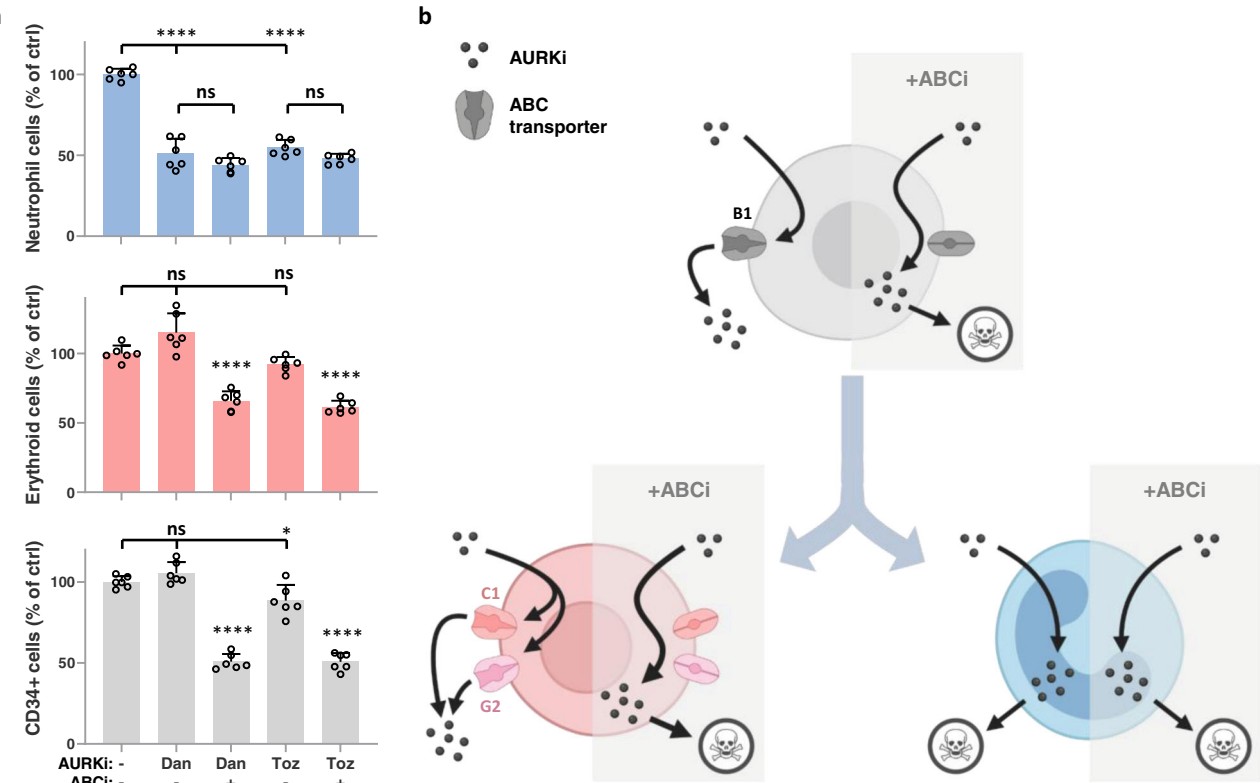

**Fig. 6 | Neutrophil-biased hematotoxicity for multiple AURKi drugs is driven by ABC transporter function. a** Human CD34+ progenitor cells were cultured for 10 days and then treated for 48 h with 100 nM danusertib, 100 nM tozasertib, or vehicle control. A combination of inhibitors of B1 (0.5 μM zosuquidar), C1 (50 μM MK-571), and G2 (0.5 μM Ko143) were also added at the time of drug treatment as indicated. Cells were harvested and cell numbers were measured by flow cytometry. Mean and SD are shown; n = 6, from 2 independent experiments. *p = 0.0188, ****p < 0.0001, ns (not significant), 1-way ANOVA with post-hoc analysis via Sidak's multiple comparisons test comparing each AURKi against vehicle control and comparing each AURKi + ABCi condition against its corresponding AURKi only condition. **b** Model of how changes in ABC transporter expression determine hematopoietic cell sensitivity to some AURK inhibitors. CD34+ progenitors cells (gray) are resistant due to B1 function. Blocking the function of B1 results in the sensitization of CD34+ progenitor cells to some AURK inhibitors due to an inability to efflux these drugs. As CD34+ progenitors cells differentiate into the erythroid lineage (red), they lose B1 function and instead rely on C1 and G2 to efflux AURK inhibitors. Blockade of C1 and G2 sensitizes erythroid cells to AURK inhibitors. In contrast, loss of ABC transporters occurs early during neutrophil differentiation (blue), rendering this hematopoietic lineage particularly sensitive to existing AURK inhibitors. Figure created with BioRender.com. Source data are provided in the Source Data file.

pathogenesis of a variety of inflammatory illnesses, including sepsis, systemic lupus erythematosus, rheumatoid arthritis, various vasculitides, and inflammatory bowel disease[53,54]. Drug developers seeking to treat these diseases by modulating neutrophil function without adversely affecting other hematopoietic cells might use differential ABC transporter expression to their advantage. To our knowledge, however, these are untested possibilities and additional studies are needed to assess their feasibility.

## Methods
### Ethical considerations
Patient samples used in this work were obtained at Massachusetts General Hospital, in compliance with Institutional Review Board (IRB)-approved protocol #2015P001859. Individual informed consent was not obtained as the risks to the anonymized donors were negligible. Donors were not participants in a clinical trial.

### Summary of cytopenia data for FDA-approved small molecule cancer therapies and AURK inhibitors
The list of all FDA-approved targeted cancer therapies was obtained from the following NCI website in early April 2020: https://www.cancer.gov/about-cancer/treatment/types/targeted-therapies/targeted-therapies-fact-sheet. Therapies that were not small molecule compounds were excluded and the FDA labels for all remaining drugs were then manually reviewed to extract the severity and rates of neutropenia, anemia, thrombocytopenia observed during the clinical trial phases of drug development. Additionally, the types of cancers included in those trials as well as whether drugs were administered as single agents or in combination with other drugs was noted. Drugs used to treat leukemias and myeloid neoplasms were then removed to avoid the confounding cytopenic effects of the underlying neoplasm. Finally, only drugs for which data were available when used as a single agent or in combination with other drugs lacking known myelosuppressive side effects were included.

A list of existing AURK inhibitors was compiled via literature review and a search for clinical trials for each AURK inhibitor was conducted on PubMed. As with the FDA-approved targeted therapies, the severity and rates of neutropenia, anemia, and thrombocytopenia were extracted and trials involving patients with an underlying myeloid neoplasm were removed.

### Cheminformatics analysis
Log P (i.e. octanol-water partition coefficient) and the ECFP_6 fingerprint for each molecule were calculated using Pipeline pilot software (version 19.1.0.1964). The molecular similarity between two molecules was measured by the Tanimoto distance of their fingerprints. Hierarchical clustering of the distance matrix was carried out using R.

## Hematopoietic CD34+ cell isolation

Mobilized peripheral blood and leukapheresis product were anonymously collected from donors undergoing stem cell mobilization at the Massachusetts General Hospital. Mononuclear cells were purified via Histopaque-1077 gradient (Sigma–Aldrich; 10771). CD34+ cells were isolated via positive magnetic bead selection using a CD34 MicroBead Kit (Miltenyi Biotec; 130-046-702) and LS columns (Miltenyi Biotec; 130-042-401), according to the manufacturer's recommendations. CD34+ purity routinely exceeded 85%, as assessed by flow cytometry. Aliquots of $3–5 \times 10^5$ cells were frozen in SFEM II medium (STEMCELL Technologies; 09655) + 10% DMSO (Sigma; 41640) + 20% fetal bovine serum (FBS; Gibco; 10082-147) using the CoolCell LX (Corning) at −80 °C, then transferred to liquid nitrogen cryogenic storage (VWR; CryoPro). Upon thawing, CD34+ cell viability was >90%, as assessed by trypan blue (Lonza; 17-942E).

## Cell culture and medium

CD34+ cells were seeded into individual wells of a 96 well tissue-culture treated flat bottom polystyrene plate ($1 \times 10^4$ cells/well in 200 μL) in SFEM II medium (STEMCELL Technologies; 09655) supplemented with 10% FBS (Gibco; 10082-147), 100 U/ml penicillin and 100 mg/ml streptomycin (Gibco; 15140-122), 12.5 μg/ml aprotinin (Sigma–Aldrich; A3428), 20 ng/ml EPO (PeproTech; 100-64), 1 ng/ml G-CSF (PeproTech; 300-23), 100 ng/ml Flt3-L (PeproTech; 300-19), 100 ng/ml TPO (PeproTech; 300-18), 50 ng/ml SCF (PeproTech; 300-07) and select EGM-2 BulletKit (Lonza; CC-4176) components (hFGF-B, VEGF, R3-IGF-1, hEGF, ascorbic acid and heparin), according to the manufacturer's instructions. Medium was changed on days 2 and 4 and every day afterwards. Cells were split 2x on both day 7 and day 10 of culture.

AZD2811 was obtained from AstraZeneca while Danusertib and Tozasertib were purchased from Selleckchem. Drugs were reconstituted in DMSO, stored frozen at −20 C, and then diluted into the above cell culture medium for use. To start drug treatment on day 10, cells were harvested into V-bottom polypropylene plates and centrifuged at room temperature for 5 min at 300xg. The supernatant was then aspirated and the cells were resuspended in medium containing drug or DMSO and transferred back to 96 well tissue-culture treated flat bottom polystyrene plates.

## Flow cytometric analysis

Harvested cells were centrifuged, resuspended in flow staining buffer composed of 1% FBS (Gibco; 10082-147), 25 mM HEPES (Thermo Fisher Scientific; 15630-080), 1 mM EDTA (Thermo Fisher Scientific; 15575-020) and 0.05% sodium azide (VWR; BDH7465-2) in Dulbecco's PBS (DPBS; Gibco; 14190-144), and filtered through 105-μm-pore nylon mesh (Component Supply; U-CMN-105-A). Extracellular antibody staining was performed for 30 min in 96-well V-bottom plates (Nunc; 249944), in 100 μl of staining buffer using the following panel: anti-CD235a-Brilliant Violet 421 (HIR2 clone; BD Biosciences; 562938; dilution: 1:100), anti-CD15-Brilliant Violet 510 (W6D3 clone; BioLegend; 323028; dilution: 1:50), anti-CD45-Brilliant Violet 570 (HI30 clone; BioLegend; 304034; dilution: 1:50), anti-CD13-Brilliant Violet 711 (WM15 clone, BioLegend 301722, dilution 1:100), anti-CD16-PE/Dazzle 594 (3G8 clone; BioLegend; 302054; dilution: 1:100), anti-CD41-PE/Cy5 (HIP8 clone; BioLegend; 303708; dilution: 1:200), anti-CD71-AF647 (CY1G4 clone; Biolegend; 334118; dilution: 1:200), anti-CD34-PE-Cy7 (561 clone; BioLegend; 343616; dilution: 1:50), Zombie NIR dye (BioLegend; 423106; dilution: 1:500), Fc Block (BioLegend; 422302; dilution: 1:20), Monocyte Blocker (BioLegend; 426103; dilution: 1:20) and Brilliant stain buffer (BD Biosciences; 566385; dilution: 1:20). Additionally, $5 \times 10^3$ counting beads (Spherotech; ACRFP-100-3) were added to each sample to enable quantification of cell numbers. After staining with surface markers, cells were washed with staining buffer

and fixed in 100 μL of 2% PFA in PBS for 30 min at 4 °C. After washing with FACS buffer, cells were centrifuged and supernatant was removed. The cell pellet was then gently vortexed (to prevent cell clumping) and permeabilized by resuspending in 150 uL of 70% ethanol for 30 min at 4 °C. Cells were washed with FACS buffer and then stained intracellularly for pHH3S10 using anti-pHH3S10-Alexa 488 (11D8 clone; BioLegend; 650804, dilution: 1:50) and Fc Block (same dilution as above). Cells were again washed and incubated overnight at 4 degrees C in 200 μL of staining buffer containing 40 ng/mL Hoechst 33342 (Life Technologies; H3570) in the dark. On the following day, cells were centrifuged, resuspended in 100 μL of staining buffer, and analyzed using the BD LSRFortessa machine and FACSDiva v8.0 software. Results were analyzed in Flowjo version 10. Representative gating strategy shown in Supplementary Fig. 7.

## Microarray analysis

Pre-processed and normalized data as well as metadata and annotation data for GSE24759 and GSE42519 were imported from the Gene Expression Omnibus (GEO) using the GEOQuery Bioconductor package[55]. Data and metadata were subset to cells of interest and annotated with the corresponding cell lineages. ProbeIDs from the microarrays were assigned to genes using the corresponding microarray annotations from GEO, preferring to use the probe with highest mean expression across all samples when multiple probes mapped to the same gene. Heatmaps were made with the pheatmap R package (Raivo Kolde, 2019, pheatmap: Pretty Heatmaps, R package version 1.0.12., https://cran.r-project.org/web/packages/pheatmap/index.html). To highlight differences between sample groups, heatmaps were row centered and scaled by removing the mean (centering) and dividing by the standard deviation (scaling) using the "scale=row" option.

## PCR analysis

Hematopoietic cells from 10 day cultures were sorted into CD34+ progenitor, CD15+ neutrophil, and CD71bright erythroid cell populations (same antibodies as above) on a Sony SH800S. Cells were then pelleted by centrifugation and RNA was isolated using an RNeasy Mini Kit (Qiagen; 74104). cDNA synthesis was done using a ReadyScript cDNA Synthesis Mix (Millipore Sigma; RDRT). Quantitative real-time PCR for ABCB1 (ThermoFisher Scientific; 4331182; Assay ID Hs00184500_m1), ABCC1 (ThermoFisher Scientific; 4331182; Assay ID Hs00219905_m1), ABCG2 (ThermoFisher Scientific; 4331182; Assay ID Hs01053790_m1), and GAPDH (ThermoFisher Scientific; 4331182; Assay ID Hs02758991_g1) was performed on a QuantStudio 7 Flex (Applied Biosystems) in a 20uL reaction and included a preamplification step (ThermoFisher Scientific; 4391128) using the TaqMan Fast Advanced Master Mix (ThermoFisher Scientific; 4444556). Threshold count values were automatically determined by the software and threshold values of B1, C1, and G2 were normalized to GAPDH thresholds for each cell population. Delta-delta CT values were then calculated relative to the CD34+ progenitor population and expressed as fold change.

## Statistical analysis

All graphs depict means ± standard deviation (s.d.) and tests for differences between groups were performed using GraphPad Prism 9 version 9.4.1 as indicated in figure legends.

## Reporting summary

Further information on research design is available in the Nature Research Reporting Summary linked to this article.

## Data availability

Previously generated microarray datasets used in this study are: GSE24759 and GSE42519. Source data are provided with this paper.

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

## Acknowledgements

This research was sponsored by funding from the US Food and Drug Administration grant HHSF223201310079C, AstraZeneca, and the Wyss Institute for Biologically Inspired Engineering (to D.E.I.) and the US National Institutes of Health (5T32CA009216-37 training grant to D.B.C.). Work by J.H. was subsidized by the Harvard Stem Cell Institute and the Center for Stem Cell Bioinformatics. The authors would like to thank M. DeLelys, A. Shay, J. Trusch, R. Nicholson, A. Alluhaydan, J. Houston, J. Patel, T. Spitzer MD, and F. Preffer PhD at Massachusetts General Hospital for their invaluable help in working with patient samples.

## Author contributions

Contribution: D.B.C. and B.A.F. participated in the design and perfor-mance of all experiments and analyzed the data working with D.E.I, who also supervised all work. R.R.P., C.K., and L.R.O. designed and per-formed experiments and analyzed data. J.H. analyzed the microarray data. J.C. performed the cheminformatics analysis. R.D., S.J.R., U.M.P., J.T., and J.U. helped design experiments and interpret data. R.P.H. pro-vided access to human cells and scientific guidance. R.P.-B. helped design experiments, interpret data, and supervised much of the work. D.B.C., B.A.F., R.R.P., C.K., A.H., and D.E.I. prepared the manuscript with input from all authors.

## Competing interests

R.D., S.J.R., U.M.P., and J.U. are employed by AstraZeneca, which is developing nanoparticle-encapsulated AZD2811. J.C. is a consultant and equity holder to Soltego, Allorion, Matchpoint, and scientific cofounder for Matchpoint, M3 bioinformatics & technology Inc. The remaining authors declare no competing interests.
