## [Peer Review File · Nature Communications]

Differential ABC transporter expression during hematopoiesis contributes to neutrophil-biased toxicity of Aurora kinase inhibitorsReviewers' Comments:

Reviewer #1 (Remarks to the Author):

Thank you for the opportunity to review this manuscript by Drs. Chou, Ingber, and colleagues. The manuscript is well-written, comprehensive, and covers an area of significant interest in oncology drug development. After reviewing the paper, I have a few comments and questions, which I will provide below:

-In the introduction, the authors posit that their goal is to first "understand the incidence of lineage-biased cytopenias induced by targeted small molecule cancer therapies..." This seems a bit too broad a goal. It appears to me that the gist of the paper is actually focused on AURK inhibitors, and particularly one drug targeting Aurora Kinase B. I would limit the scope of your goals in the introduction unless you plan to do the analyses you have done with AZD2811 with a broad range of other small molecule inhibitors.

-The authors go on the review a variety of small molecule inhibitors from the NCI website. However, to my knowledge, no aurora kinase inhibitor has yet been approved. This seems incongruent. If the goal is to define the unique aspects of all small molecule inhibitors, why focus on cytopenia-inducing potential of only FDA-approved drugs? My own suggestion would be to focus instead on this potential for the variety of aurora kinase inhibitors which have been tested in clinical studies to date. There are a variety of reasons why a small molecular inhibitor may cause cytopenia, and many of these probably are not related to the ABC transporter, or perhaps are but there is differential expression. I worry this just conflates phenomena and confuses the picture.

-Why were drugs treating hematologic malignancies removed from any analyses. Although I agree that signal for cytopenia may be difficult to capture with these diseases, hematologic DLTs and >G2 toxicities can also be captured.

-I noticed that the analysis looked particularly at danusertib and tozasertib. These, from my understanding, are pan aurora kinase inhibitors. Would it not be interesting to also compare the more specific AURKB inhibitor, AZD2811, to perhaps a specific AURKA inhibitor, to determine whether there are differential effects in terms of mechanisms underlying promotion of cytopenia? Otherwise, it may be difficult to understand whether the effect is specific to AURKB inhibitors, or is a broader effect applicable to all AURK inhibitors.

-The finding of the toxicity focused on the neutrophil lineage is indeed intriguing, and I commend the authors on their work to try and determine why this may be. Could one knock out ABCB1, C1, or G2, respectively to see if one can render CD34 progenitor cells, etc, susceptible to AURKB inhibition and cell death, as further evidence for your very reasonable theory. Or is this not feasible for some technical reason? You use inhibitors instead, and I am wondering why. You do mention in your Discussion that there is a B1 knockout model available.

-It does appear that there is a preponderance of neutrophil lineage toxicity with these agents, but there is also a signal in some trials for severe thrombocytopenia and anemia. For example, with alisertib, the AURKA inhibitor, other G3 or higher cytopenias have been seen in clinical trials of mainly solid tumor patients (Falchook et al, JAMA Oncology 2019, Strati et al, Haematologica 2020, Beltran et al, Clin Cancer Res 2019, etc). Could this be through a separate mechanism of action? I would suggest commenting a bit more on other aurora kinase inhibitors studied in clinical trials.

-In your discussion, you state that " ... the results clearly illustrate how the expression patterns of these drug efflux pumps play a clinically relevant role in hematotoxicities experienced by patients receiving certain types of cancer therapies." This comment is too strong based on the data presented. Your data is very intriguing and suggestive of a connection, but I am not certain they clearly establish this as the main and only cause of neutropenia.

Reviewer #2 (Remarks to the Author):

In this manuscript, Chou, Furlong and Posey et al use donor-derived CD34+ cells to understand the mechanism behind aurora kinase inhibitor-induced selective neutropenia. The authors propose that downregulation of ABC efflux transporters, particularly ABCG2, during neutrophil differentiation accounts for their enhanced sensitivity to AURKi. The effort and in general, large sample numbers are commendable. However, given the proposed mechanism of transporter-mediated efflux, it raises the question of whether neutrophil sensitivity to AURKi is attributable to the drugs' biological activity or rather a structural feature of these drugs. Are there structural features of AZD2811, danusertib, tozasertib, AMG-900, AT9283, etc that are associated with more severe neutropenia? Are those structural features found in other classes of drugs that induce neutropenia?

Figure 1B: Can a statistical test be applied to determine whether this enrichment of Aurora kinase inhibitors in severe neutropenia is significant?

Figure 2B: Pulse label proliferating cells with eg. EdU to indicate that cells have progressed through/stalled at mitosis.

Figure 2B and 2C: Is the Hoechst 33342 labelling from the same experiment (ie. from fixed and permeabilized cells?) If Figure 2B was data from live cells, were the cells treated with verapamil/reserpine to inhibit efflux of the DNA dye?

Figure 5A: Why is the sample number so low? Was it not possible to combine the two independent experiments?

Figure 5A: How does efflux transporter expression compare in the lymphoid and monocytic lineages?

Figure 6A: There is a population that has $>2n$ Hoechst staining in the cells treated with the ABC inhibitors, especially in the cells treated with all the inhibitors. Can the authors comment on what this population is? It is presumed that doublets were excluded in the gating strategy, but the authors should provide the flow cytometry gating strategy for their experiments.

RESPONSE TO REVIEWERS

(Chou et al., NCOMMS-21-41388)

Reviewer #1:

1. In the introduction, the authors posit that their goal is to first "understand the incidence of lineage-biased cytopenias induced by targeted small molecule cancer therapies..." This seems a bit too broad a goal. It appears to me that the gist of the paper is actually focused on AURK inhibitors, and particularly one drug targeting Aurora Kinase B. I would limit the scope of your goals in the introduction unless you plan to do the analyses you have done with AZD2811 with a broad range of other small molecule inhibitors.

We agree with the Reviewer and have reworded that paragraph in the Introduction to more accurately reflect the focus of the manuscript.

2. The authors go on the review a variety of small molecule inhibitors from the NCI website. However, to my knowledge, no aurora kinase inhibitor has yet been approved. This seems incongruent. If the goal is to define the unique aspects of all small molecule inhibitors, why focus on cytopenia-inducing potential of only FDA-approved drugs? My own suggestion would be to focus instead on this potential for the variety of aurora kinase inhibitors which have been tested in clinical studies to date. There are a variety of reasons why a small molecular inhibitor may cause cytopenia, and many of these probably are not related to the ABC transporter, or perhaps are but there is differential expression. I worry this just conflates phenomena and confuses the picture.

We acknowledge the Reviewer's point that this is not a main focus of the paper and may be confusing, so we have moved the data to **Supplementary Fig. 1**. We also agree with the Reviewer that there are many factors that contribute to drug-induced cytopenias and now elaborate on some of those in the Discussion.

We still include the comparison to FDA-approved small molecule cancer therapies, however, because we believe it provides useful context. When we first realized the neutropenia bias of AZD2811 extended to all AURK inhibitors, we naturally wondered about the extent to which other targeted cancer drugs would behave similarly and we believe future readers of our manuscript may have a similar question. The set of FDA-approved small molecule cancer therapies is a useful reference because it contains chemically diverse compounds for which high quality clinical information describing their effects in patients is readily available. Furthermore, as requested by Reviewer 2, we now include new contingency tables and statistical analyses in **Supplementary Fig. 1** confirming that the distribution of cytopenias is significantly different between the AURK inhibitors and the FDA-approved small molecule reference dataset.

3. Why were drugs treating hematologic malignancies removed from any analyses. Although I agree that signal for cytopenia may be difficult to capture with these diseases, hematologic DLTs and >G2 toxicities can also be captured.

We actually only removed the drugs/studies involving acute leukemia or a myeloid neoplasm. The reason for this is that those diseases either intrinsically cause severe cytopenias (and thus patients would be cytopenic regardless of drug treatment) or they represent disorders in which the myeloid, erythroid, and megakaryocytic precursors are neoplastic and thus not representative of normal hematopoietic biology. In both cases, the peripheral cytopenia data is confounded by the underlying disease and including it would be misleading. However, other hematopoietic malignancies, such as lymphomas, are included in the analysis because they do not have the same biology and thus do not confound the cytopenia results in the same way. This is now clarified in the text.

4. I noticed that the analysis looked particularly at danusertib and tozasertib. These, from my understanding, are pan aurora kinase inhibitors. Would it not be interesting to also compare the more specific AURKB inhibitor, AZD2811, to perhaps a specific AURKA inhibitor, to determine whether there are differential effects in terms of mechanisms underlying promotion of cytopenia? Otherwise, it may be difficult to understand whether the effect is specific to AURKB inhibitors, or is a broader effect applicable to all AURK inhibitors.

We agree that it is possible that AURK specificity could influence the cytopenic effects of this class of drugs and is worth investigating in future studies. However, in clinical use, the plasma levels of even the most selective AURKA inhibitor (MLN8237) routinely reaches concentrations at which both AURKA and AURKB are inhibited (PMIDs: 22016509, 22753585, 22767670, 22772063, etc). Thus, the neutropenic bias observed in patients is unlikely to be mechanistically related to AURK isoform specificity. In any case, the main message of our manuscript is that ABC transporter function explains how selective lineage toxicity arises in the setting of a general mitotic inhibitor. Because all AURK inhibitors are mitotic inhibitors, the specific AURK isoform targeted by the drug does not significantly alter the message.

In addition, we now include a new **Supplementary Fig. 2**, which lists the IC50 values for the various AURK inhibitors for AURKA, AURKB, and AURKC to illustrate their selectivity for the different isoforms along with their cytopenic effects. Comparing the 3 AURKA-selective compounds (tozasertib has the 2nd highest selectivity ratio for AURKA) to AZD2811 does not suggest any association between AURK isoform selectivity and cytopenia.

5. The finding of the toxicity focused on the neutrophil lineage is indeed intriguing, and I commend the authors on their work to try and determine why this may be. Could one knock out ABCB1, C1, or G2, respectively to see if one can render CD34 progenitor cells, etc, susceptible to AURKB inhibition and cell death, as further evidence for your very reasonable theory. Or is this not feasible for some technical reason? You use inhibitors instead, and I am wondering why. You do mention in your Discussion that there is a B1 knockout model available.

We use the commercially available ABC transporter inhibitors in our paper because they are validated to block ABC transporter function and thus routinely used and widely accepted in the field. Furthermore, we can add them specifically for the duration of drug treatment and easily use them in combination. One could, in theory, knock out ABCB1, C1, and G2 using gene editing methods but doing this at high enough efficiency in primary human CD34+ cells would require significant optimization,

especially if multiple simultaneous gene knockouts are desired. Additionally, the process of gene editing and removing these genes may influence cellular differentiation in an unexpected manner.

The B1 knockout is a mouse model whereas we focus on human cells because of their clinical relevance in this study. Importantly, the neutrophil sensitivity to AZD2811 seen human patients is not observed in mice (internal data; AstraZeneca).

6. It does appear that there is a preponderance of neutrophil lineage toxicity with these agents, but there is also a signal in some trials for severe thrombocytopenia and anemia. For example, with alisertib, the AURKA inhibitor, other G3 or higher cytopenias have been seen in clinical trials of mainly solid tumor patients (Falchook et al, JAMA Oncology 2019, Strati et al, Haematologica 2020, Beltran et al, Clin Cancer Res 2019, etc). Could this be through a separate mechanism of action? I would suggest commenting a bit more on other aurora kinase inhibitors studied in clinical trials.

We also note the presence of G3+ anemia and/or thrombocytopenia with the AURK inhibitor family, though to a lesser degree than G3+ neutropenia, and this is seen in the locations of the AURKi datapoints in the original **Fig. 1B** (now **Supplementary Fig. 1**). We interpret this as being congruent with our findings. We show that non-neutrophil hematopoietic lineages are resistant to these drugs due to ABC transporter expression but high enough concentrations of these drugs can overwhelm their efflux capacity. In clinical practice, the pharmacokinetics of these drugs are likely to result in peak drug concentrations that exceed, at least temporarily, the efflux capacity of the cells. Furthermore, this will vary significantly between patients in a manner that is dependent on an individual patient's weight, liver and kidney function (drug metabolism and clearance), and other physiologic factors. Additionally, patients may be taking medications or supplements that interfere with ABC transporter function, making those patients more susceptible to non-neutrophil cytopenias.

Thus, our findings provide insight into how preferential neutropenia can arise with these drugs and also provide a potential framework for thinking about why a patient might experience significant cytopenias in other lineages. As per the Reviewer's suggestion, we now elaborate on these points in the Discussion.

7. In your discussion, you state that " ... the results clearly illustrate how the expression patterns of these drug efflux pumps play a clinically relevant role in hematotoxicities experienced by patients receiving certain types of cancer therapies." This comment is too strong based on the data presented. Your data is very intriguing and suggestive of a connection, but I am not certain they clearly establish this as the main and only cause of neutropenia.

We did not mean to suggest that our findings are the main and only cause of neutropenia but rather that they play a role in the hematotoxicities experienced by patients. This sentence was removed in the course of revising the Discussion.

Reviewer #2:

1. In this manuscript, Chou, Furlong and Posey et al use donor-derived CD34+ cells to understand the mechanism behind aurora kinase inhibitor-induced selective neutropenia. The authors propose that downregulation of ABC efflux transporters, particularly ABCG2, during neutrophil differentiation accounts for their enhanced sensitivity to AURKi. The effort and in general, large sample numbers are commendable. However, given the proposed mechanism of transporter-mediated efflux, it raises the question of whether neutrophil sensitivity to AURKi is attributable to the drugs' biological activity or rather a structural feature of these drugs. Are there structural features of AZD2811, danusertib, tozasertib, AMG-900, AT9283, etc that are associated with more severe neutropenia? Are those structural features found in other classes of drugs that induce neutropenia?

We thank the Reviewer for commending the effort we undertook to elucidate how a significant dose-limiting toxicity in cancer therapies arises from changes in ABC transporter expression during hematopoietic development.

To answer the Reviewer's questions, we have consulted with a cheminformatics expert in an attempt to define structural features associated with neutropenia or correlated with greater neutropenia incidence. This analysis is now provided in **Supplementary Fig. 3** and is elaborated upon in both the Results and the Discussion. The findings demonstrate that existing AURKi drugs are structurally diverse and do not cluster together when compared to the FDA-approved small molecule comparison dataset. Additionally, the neutropenia-biased drugs show no tendency to cluster together when compared to drugs without a neutropenia bias. Analysis of the overall lipophilicity of these compounds also shows no difference among these different groups of molecules. Taken together, these findings show that the neutropenia-bias of these drugs is not explained by analyzing chemical structural features, at least with current cheminformatics methods.

Also, these findings remain in agreement with our paper's central message, which is that the neutropenia bias of AURK inhibitors is not attributable to their biological activity, per se, but rather their nature as ABC transporter substrates (exactly as the Reviewer has suggested). A significant body of research has gone into trying to predict what structural features of compounds make them good substrates for ABC transporters. The field has shown that these pumps are capable of effluxing a diverse array of molecules and identifying substrates of ABC transporters remains a difficult problem in drug development that relies on empirical methods more than in silico prediction. This important piece of context is now included in the Discussion. Accordingly, our main conclusion is supported by multiple lines of evidence, including dose titration studies, recovery time course studies, gene expression by PCR and microarray analysis, and functional proof via efflux pump inhibition. We thank the Reviewer for suggesting that we pursue this analysis as these findings greatly strengthen our manuscript.

2. Figure 1B: Can a statistical test be applied to determine whether this enrichment of Aurora kinase inhibitors in severe neutropenia is significant?

We now include contingency tables and statistical analyses for the comparison of incidences of neutropenia vs anemia and neutropenia vs thrombocytopenia, showing that the distribution of these side effects is significantly different between AURK inhibitors and existing FDA-approved small molecule

cancer therapies. As per both Reviewers' comments, this data is now included as revised **Supplementary Fig. 1**.

3. Figure 2B: Pulse label proliferating cells with eg. EdU to indicate that cells have progressed through/stalled at mitosis.

Pulse labeling cells with EdU would identify those cells that are actively synthesizing DNA during the period of EdU inclusion, but this is not required for our purposes. Instead, we measure the cell cycle state by DNA content as this is consistent with prior literature studying the effect of AURKi drugs in vitro and is also commonly used in the hematopoietic literature (PMID: 17495131, 19366807, 32877675, etc). In an actively proliferating culture of primary hematopoietic cells, the buildup of cells with 4N DNA content is acceptable for demonstrating mitotic blockade (PMID: 19366807). In addition, because we combine DNA content with co-staining for pH3S10 (which labels cells actively undergoing mitosis at the time of analysis), we directly measure mitotic blockade, which is an appropriate basis for our conclusions.

4. Figure 2B and 2C: Is the Hoechst 33342 labelling from the same experiment (ie. from fixed and permeabilized cells?) If Figure 2B was data from live cells, were the cells treated with verapamil/reserpine to inhibit efflux of the DNA dye?

The Hoechst labeling is from the same experiment and is done on fixed+permeabilized cells. This is now clarified in the Methods.

5. Figure 5A: Why is the sample number so low? Was it not possible to combine the two independent experiments?

We originally chose to show one experiment because the specific fold changes are different between experiments while the trends are the same. In any case, as the Reviewer's interest in seeing all the data is understandable, we have updated **Fig. 5a** so that it now includes results for the two independent experiments.

6. Figure 5A: How does efflux transporter expression compare in the lymphoid and monocytic lineages?

Our in vitro suspension cultures are best suited for modeling neutrophil and erythroid development and less appropriate for modeling lymphoid and monocytic differentiation because of the cytokine cocktail we provide. We focus on those lineages because neutropenia and anemia are clinically dominant measures of hematotoxicity in drug development.

However, the microarray data in **Fig. 5b** illustrates the granulocyte-monocyte progenitors that give rise to monocytes have low expression of B1, C1, and G2, similar to the more differentiated neutrophils. And as we mention in the Discussion, human monocytes are known to have low expression of B1. Mapping efflux pump expression amongst the diverse populations of human lymphocytes, including B cells, T cells, and NK cells at varying stages of differentiation, is a worthy goal but beyond the

scope of our current work. We focus on clinical toxicities in our work and detailed analysis of lymphoid cells is not routinely performed as part of hematotoxicity monitoring in patients.

The Reviewer's question raises the related issue of whether megakaryocytes express ABC transporters. Thrombocytopenia is another critical hematotoxicity readout during new drug development. Our multilineage differentiation cultures do not accurately represent megakaryocyte biology, so we reanalyzed the microarray data and now show in **Supplementary Fig. 3** that megakaryocytes retain ABCB1 expression. Consistent with our manuscript's central message, thrombocytopenia is an infrequent occurrence with AURK inhibitors.

7. Figure 6A: There is a population that has >2n Hoechst staining in the cells treated with the ABC inhibitors, especially in the cells treated with all the inhibitors. Can the authors comment on what this population is? It is presumed that doublets were excluded in the gating strategy, but the authors should provide the flow cytometry gating strategy for their experiments.

The Reviewer makes an insightful observation (we assume the Reviewer means the small >4N population and not the 4N population that is normal and expected due to DNA synthesis as cells proliferate). That small population represents polyploid cells and is a known phenomenon that occurs with mitotic inhibitors; it is thought to be one of the pathways leading to cancer cell death. We include an illustrative figure on the right for clarity.

This population is most clearly seen in **Fig. 6A** because the x-axis is linear scale to better highlight this exact point. Furthermore, one can see that this >4N population appears in neutrophil lineage cells upon treatment with AZD2811 but not in erythroid cells unless ABC transporter function is also blocked, supporting our manuscript's conclusions. As requested by the Reviewer, we now comment on this in the Results when discussing **Fig. 6A** and provide our gating strategy in **Supplementary Fig. 7**.

REVIEWER COMMENTS

Reviewer #1 (Remarks to the Author):

Thank you for your revisions and responses.

Reviewer #2 (Remarks to the Author):

- The addition of cheminformatics to this version of the manuscript was really interesting but it does complicate the authors' messaging in this manuscript. It raises the possibility that the neutrophil lineage fails to efflux many drugs (including drugs for conditions other than cancer) and that many drugs should have a neutropenic effect. Yet, as supplementary Figure 1 would suggest, this is not the case. Doesn't this therefore suggest the mechanism of AURK sensitivity involves, to a certain extent, the drugs' biological activity?

- Supplementary Figure 1: Comment on the other 6 drugs in which N>A or N>T. Do these drugs target diverse cellular processes, or are mitotic inhibitors overrepresented in this group? Since there are only 6 non-AURKi drugs, it might be helpful to list them.

- Explain why the study chose to focus on AZD2811. Define the profile similarity that was used to justify examining danusertib and tozasertib in Supplementary Fig 6 (as opposed to eg. AT9283 and PF-03814735 which to this reviewer, seem to have a similar cytopenia profile).

Just a comment that this presents a precision medicine opportunity, in which ABCB1/ABCC1/ABCG2 polymorphisms can be used to inform the use of AURK inhibitors in cancer patients.

Reviewer #3 (Remarks to the Author):

The selection of approved drugs from NCI was properly performed, they took into account biological knowledge relevant to the study, rather than comparing it with the entire approved drug set. This should be emphasized throughout the text, to make clear that the selection of molecules is reasonable for the study and support the conclusions reached.

The authors choose ECFP to represent their molecules. Any conclusion should be referred to the scope of this representation. Moreover, the authors should state the reason for their selection and briefly describe the characteristics and scope of this fingerprint.

* In the results section the authors stated, "We also analyzed the chemical structures of the AURK inhibitors and FDA-approved compounds to determine if there might be structural features that correlate with induction of neutropenia."

I suggest using “identify” instead of “determined” and “associated” instead of “correlate”. Correlation is often used to refer only to a linear relationship between two variables.

* “clustering of the molecular fingerprints”

what is clustered are the molecules...based on the molecular representation employed, in this case, molecular fingerprints.

“clustering of the molecular fingerprints for all of these compounds did not reveal any common structural features as the AURK inhibitors did not cluster together and thus, they were as chemically diverse as the FDA-approved drugs”

Clustering analysis based on ECFP will show whether the molecules populate the same region of the chemical space based on ECFP. Also, strictly speaking, the sentence “did not reveal any common structural features” is not accurate. The conclusions are delimited by the fingerprint employed.

“therefore as structurally diverse as drugs” since similarity as well as diversity depend on the molecular representation employed, the conclusions should be bounded by the representation employed.

The sentence “neutrophil-biased toxicity profile of AURK inhibitors is not generally true...” is not clear.

In the figure caption of supplementary Fig 3. Change “similarly diverse” to “equally diverse” or something alike, as similarity and diversity are opposite terms commonly employed in chemoinformatics.

RESPONSE TO REVIEWERS

(Chou et al., NCOMMS-21-41388B)

Reviewer #2:

1. *The addition of cheminformatics to this version of the manuscript was really interesting but it does complicate the authors' messaging in this manuscript. It raises the possibility that the neutrophil lineage fails to efflux many drugs (including drugs for conditions other than cancer) and that many drugs should have a neutropenic effect. Yet, as supplementary Figure 1 would suggest, this is not the case. Doesn't this therefore suggest the mechanism of AURK sensitivity involves, to a certain extent, the drugs' biological activity?*

We agree that the mechanism of action for all drugs, including AURK inhibitors, is an important factor that determines their cytopenic profile. For example, the graphs below show that PARP inhibitors display preferential anemia, suggesting that PARP activity is an important part of erythroid development. This is supported by published data showing that PARP2-deficient mice are anemic (PMID: 25501596).

Thus, as the Reviewer suggests, it is possible that there is at least some aspect of the neutropenia-bias that is related to the underlying mechanism of AURK inhibition. While it is unclear to us how mitotic blockade would be cell-type specific, it is interesting to speculate that mitotic kinases may be used differently by neutrophils compared to other hematopoietic lineages. However, this has not been shown. In response to this insightful

comment, we now discuss this point in the 3rd paragraph of the Discussion. We further emphasize in the 4th paragraph of the Discussion that ABC transporters are just one contributing factor to patient cytopenias by stating "...our results provide insight into one aspect of a larger complex set of interactions".

2. *Supplementary Figure 1: Comment on the other 6 drugs in which N>A or N>T. Do these drugs target diverse cellular processes, or are mitotic inhibitors overrepresented in this group? Since there are only 6 non-AURKi drugs, it might be helpful to list them.*

There are 14 non-AURKi drugs that are N>A or N>T (listed in 2 columns on the right), which target a variety of cellular processes including receptor tyrosine kinases, intracellular signaling kinases, mTOR, cyclin-dependent kinases, and hormone receptors. As per the Reviewer's suggestion, we now include a new **Supplementary Table 1** that lists the AURK inhibitors and all FDA-approved targeted cancer therapies which passed our filter along with their corresponding cytopenia data, PubChem IDs, and SMILES representations as a resource for other researchers.

N > A	N > T
abemaciclib	abemaciclib
acalabrutinib	avapritinib
bexarotene	bexarotene
bortezomib	copanlisib
cabozantinib	crizotinib
copanlisib	duvelisib
crizotinib	entrectinib
duvelisib	everolimus
everolimus	larotrectinib
palbociclib	palbociclib
ribociclib	ribociclib
romidepsin	sunitinib
sunitinib	talazoparib
zanubrutinib	zanubrutinib

- 3. Explain why the study chose to focus on AZD2811. Define the profile similarity that was used to justify examining danusertib and tozasertib in Supplementary Fig 6 (as opposed to eg. AT9283 and PF-03814735 which to this reviewer, seem to have a similar cytopenia profile).***

We focused on AZD2811 because these studies arose from an existing collaborative project with AstraZeneca in which we were studying the side effect profile of this compound in a different context. Thus, we had existing data that showed we could recapitulate its neutrophil-biased toxicity profile in vitro. We chose danusertib and tozasertib simply as additional representative members of the AURKi family with varying potencies against AURKA and AURKB. This is now clarified in the text.

- 4. Just a comment that this presents a precision medicine opportunity, in which ABCB1/ABCC1/ABCG2 polymorphisms can be used to inform the use of AURK inhibitors in cancer patients.***

We thank the Reviewer for making this interesting point and we now share this observation in the revised Discussion.

Reviewer #3:

- 1. The selection of approved drugs from NCI was properly performed, they took into account biological knowledge relevant to the study, rather than comparing it with the entire approved drug set. This should be emphasized throughout the text, to make clear that the selection of molecules is reasonable for the study and support the conclusions reached.***

We have added clarifying text in the Results and the Discussion sections, as per the Reviewer's suggestion.

- 2. The authors choose ECFP to represent their molecules. Any conclusion should be referred to the scope of this representation. Moreover, the authors should state the reason for their selection and briefly describe the characteristics and scope of this fingerprint.***

We have added text describing ECFP and its characteristics, as suggested by the Reviewer.

- 3. In the results section the authors stated, "We also analyzed the chemical structures of the AURK inhibitors and FDA-approved compounds to determine if there might be structural features that correlate with induction of neutropenia." I suggest using "identify" instead of "determined" and "associated" instead of "correlate". Correlation is often used to refer only to a linear relationship between two variables.***

We made the changes suggested by the Reviewer.

- 4. "clustering of the molecular fingerprints" what is clustered are the molecules...based on the molecular representation employed, in this case, molecular fingerprints..."clustering of the molecular fingerprints for all of these compounds did not reveal any common structural features***

as the AURK inhibitors did not cluster together and thus, they were as chemically diverse as the FDA-approved drugs” Clustering analysis based on ECFP will show whether the molecules populate the same region of the chemical space based on ECFP. Also, strictly speaking, the sentence “did not reveal any common structural features” is not accurate. The conclusions are delimited by the fingerprint employed... “therefore as structurally diverse as drugs” since similarity as well as diversity depend on the molecular representation employed, the conclusions should be bounded by the representation employed.

We accept these valid points and have changed the wording of this section to more accurately reflect the specifics of our ECFP-based analysis. It now reads:

“To do this, we performed molecular fingerprinting analysis using extended-connectivity fingerprint of diameter 6 (ECFP_6), which calculates a numerical fingerprint for each drug based upon all possible substructures within the molecule with a maximum radius of 3 bonds (and thus a diameter of 6)^{14,15}. This is a widely used fingerprinting method that has proven utility in a variety of scientific applications¹⁵. Clustering of the molecules based on their ECFP_6 fingerprint showed that the AURK inhibitors did not cluster separately from the FDA-approved drugs (Supplementary Fig. 3a). Furthermore, neutropenia-biased drugs did not cluster separately from drugs without preferential neutropenia (Supplementary Fig. 3a). Thus, according to their ECFP_6 representation, all of these compounds distribute over the same region of chemical space. Analysis of the overall lipophilicity of the compounds also showed no significant difference between AURK inhibitors and the FDA-approved compounds or between drugs with and without a neutropenia-bias (Supplementary Fig. 3b, c). Taken together, these findings show the neutrophil-biased toxicity profile of AURK inhibitors is not a general property of small molecule cancer drugs and that this neutropenia-bias occurs in the context of significant chemical structural diversity.”

5. The sentence “neutrophil-biased toxicity profile of AURK inhibitors is not generally true...” is not clear.

This sentence was removed during revision. The section now reads:

“Taken together, these findings show the neutrophil-biased toxicity profile of AURK inhibitors is not a general property of small molecule cancer drugs and that this neutropenia-bias occurs in the context of significant chemical structural diversity.”

6. In the figure caption of supplementary Fig 3. Change “similarly diverse” to “equally diverse” or something alike, as similarity and diversity are opposite terms commonly employed in chemoinformatics

We made the suggested change.